# Stable Isotope Analysis of Residual Pesticides via High Performance Liquid Chromatography and Elemental Analyzer–Isotope Ratio Mass Spectrometry

**DOI:** 10.3390/molecules27238587

**Published:** 2022-12-06

**Authors:** Hee Young Yun, Eun-Ji Won, Jisoo Choi, Yusang Cho, Da-Jung Lim, In-Seon Kim, Kyung-Hoon Shin

**Affiliations:** 1Institute of Marine and Atmospheric Sciences, Hanyang University, Ansan 15588, Republic of Korea; 2Department of Agricultural Chemistry, Chonnam National University, Gwangju 61186, Republic of Korea

**Keywords:** compound-specific isotope analysis, pollutant, agricultural application, soil, HPLC, SPE extraction

## Abstract

To broaden the range of measurable pesticides for stable isotope analysis (SIA), we tested whether SIA of the anthranilic diamides cyantraniliprole (CYN) and chlorantraniliprole (CHL) can be achieved under elemental analyzer/isotope ratio mass spectrometry with compound purification in high-performance liquid chromatography (HPLC). Using this method, carbon isotope compositions were measured in pesticide residues extracted from plants (lettuce) grown indoors in potting soil that were treated with 500 mg/kg CHL and 250 mg/kg CYN and were followed up for 45 days. Our results show that the CYN and CHL standard materials did not have significant isotope differences before and after clean-up processing in HPLC. Further, when applied to the CYN product and CHL product in soil, stable isotope differences between the soil and plant were observed at <1.0‰ throughout the incubation period. There was a slight increase in the variability of pesticide isotope ratio detected with longer-term incubation (CHL, on average 1.5‰). Overall, we measured the carbon isotope ratio of target pesticides from HPLC fraction as the purification and pre-concentration step for environmental and biological samples. Such negligible isotopic differences in pesticide residues in soils and plants 45 days after application confirmed the potential of CSIA to quantify pesticide behavior in environments.

## 1. Introduction

Pesticides are used to prevent crop damage from pest insects and pathogens and to prolong the storage lives of agricultural products. However, persistence in pesticide-contaminated soils and repeated use can lead to an increase in the unintentional buildup of pesticide(s) residues that adversely affect non-target organisms and cause insecticide resistance, and negatively impact the environment and human health [1]. Many countries are seeking sustainable agriculture and farming practices by decreasing pesticide applications and by enforcing safe (or maximum acceptable) levels of pesticide concentrations detected in crops and agricultural products [2]. In this view, developing analytic techniques for pesticides in environmental and crop samples has received our attention. Current analytical approaches have focused on identifying pesticide compounds with a wide range of differences in physiochemical characteristics, polarity and thermal stability [3,4]. Although determining pesticide presence and their concentration from a complex matrix is an important mission for environmental pollution monitoring and for food safety issues, this concentration-based approach has difficulties in tracing the sources of pesticide contamination, where pesticides are directly given or sprayed in the environments [3].

Compound-specific stable isotope analysis (CSIA) is a standard way of pollutant risk assessment [3,5] for identifying their behavior and distribution patterns. It ultimately allows for characterizing pollutant sources (i.e., point- vs. nonpoint-contamination source) and their relative contributions. This applicability is based on the fact that naturally occurring stable isotope compositions of elements (e.g., ^13^C/^12^C) in pesticide residues are closely related to stable isotope ratios of their source (parental compounds). Further, the carbon isotope ratio of the pesticide is changed only negligibly or slightly by plant absorption [6,7] and abiotic degradation processes (e.g., photolysis and hydrolysis) [8]. In contrast, soil type (sterile vs. non-sterile soil) and biodegradation processing fairly increase carbon isotope ratios in pesticides <13.5‰, e.g., throughout the incubation period (e.g., from days to months) [7]. Microorganisms tend to metabolize chemical substances with lighter isotopes of elements (e.g., ^12^C) because of the lower energy required for breaking the bonds with the lighter isotope (i.e., the kinetic isotope effect). Thus, biodegradation leads to more compounds with heavier isotopes of elements (e.g., ^12^C) remaining in the substrates than those with the lighter isotope, increasing the stable isotope variables [8,9]. Such distinctive isotopic variability in pesticides from abiotic or biodegradation processing helps detect the dynamic behavior of pesticide molecules in environments [3,6,10].

CSIA [3,6,10,11,12] is selectively accessible for several pesticides (insecticides, herbicides, and fungicides) (Figure 1) and pesticide metabolites, e.g., aminomethylphosphonic acid from glyphosate and desphenylchloridazon from chloridazon [13].

Particularly, carbon is the most common element in pesticide substances and their metabolite compounds rather than other elements such as nitrogen [14]. The isotope ratios are mainly measured by an isotope ratio mass spectrometer (IRMS) system coupled to chromatographic systems such as gas chromatography (GC) and liquid chromatography (LC) (called GC–IRMS and LC–IRMS, respectively) depending on the characteristics of organic compounds such as polarities and molecular weight. As shown in Figure 1, GC–IRMS has generally been used for measuring the isotope composition of non-polar and volatile organic compounds, whereas LC–IRMS for polar and less volatile compounds (summarized in [3,5]). Additionally, an elemental analyzer (EA) directly connected to the IRMS system is capable of measuring solids or liquids (both types found in pesticide compounds) and is not restricted by characteristics of organic compounds such as polarities and molecular weights. EA–IRMS is generally used to determine the mass fractions of carbon, nitrogen, hydrogen, and sulfur of any organic compounds in diverse research fields (e.g., [3,15]). However, EA–IRMS does not inherently perform chromatographic separation for diverse organic compounds. To counter these issues, offline high-performance liquid chromatography (HPLC) methods were preceded prior to EA–IRMS (referred to as HPLC/EA–IRMS or the ‘offline’ CSIA method). HPLC/EA–IRMS were applied for measuring carbon isotopes in several biomolecules such as porphyrin [16] and amino acids [17,18] but rarely applied for pollutants (including pesticides).

Residual pesticides are commonly low in crops relative to topsoil and the CSIA approach in crop and environmental samples should collect a measurable amount of target analytes via significant extraction and pre-concentration steps. Moreover, traditional EA–IRMS systems require much higher amounts of analytes that produce reliable isotope variables compared to a GC–IRMS system. That is, the analyte mass required for EA–IRMS is approximately 20 µgC (or 50 µgN) [16,18,19], while the analyte mass for GC–IRMS via direct injection on the column is around 0.2 to 0.02 µgC (or 0.5 to 0.05 µgN). Fortunately, an improved capacity of the HPLC fraction collector (available injection volume up to 1 mL) with automated chromatographic separation might reduce the workload needed to detect target analytes over the detection limit. Additionally, practical HPLC/EA–IRMS methods should include the efficient removal of carbon-containing solvents (e.g., acetonitrile, dichloromethane, and methanol), since EA–IRMS does not have the ability to separate a pesticide compound from solvents based on molecular characteristics. Otherwise, the carbon isotope ratios of target analytes containing solvents might be shifted by carbon-containing solvents. Method optimization (i.e., extraction and clean-up procedure for complex matrices in soil and crop samples) of the EA–IRMS analysis is needed for broadening the measurable range of pesticides and for enhancing the CSIA applicability in environmental and agricultural issues.

The diamide pesticides such as cyantraniliprole (CYN) and chlorantraniliprole (CHL) used in this study are common due to their promising pest management effects via selective, uncontrolled calcium homeostasis in insects [20]. These pesticides were not applied in previous CSIA work, which has focused only on concentration determination in which CHL [21,22] and CYN [23] were used as parental compounds as well as metabolites [1]. Here, we sought to investigate an HPLC/EA–IRMS method for a CSIA approach in residual pesticides. First, the range of isotope ratio variability based on pesticide standards analyzed in EA–IRMS was compared before and after the analysis of HPLC fractions to verify the effects of isolation and pre-concentration procedures on pesticide isotope ratios. We also demonstrate the potential utility of this method by evaluating isotope compositions of the pesticide residue extracted from the soil and moved to lettuce in planted mesocosm set-ups with comparisons to the pesticide source. This CSIA approach will serve to monitor more diverse pesticide residues in the environment.

## 2. Results and Discussion

To our knowledge, this is the first study to measure stable isotope ratios of pesticides in EA–IRMS. Firstly, we investigated the possible uncertainties with isotopic measurements for pesticides CHL and CYN in HPLC/EA–IRMS. By combining a typical sorbent extraction method and chromatographic separations (time-based approach in HPLC chromatograms) with injection volume <200 μL, we collected considerable amounts of analytes (>3 nA particularly soil extract, corresponding to 0.15 mgC for CHL and 0.17 mgC for CYN) for obtaining reliable carbon isotopic compositions. Our HPLC/EA–IRMS approach helps to analyze stable isotope ratios of pesticides that had not been previously carried out, i.e., CHL and CYN in other studies, and this technique might broaden the range of measurable pesticides if LC–IRMS or GC–IRMS is not available.

### 2.1. Error Evaluation during Isotope Measurement

The effects from the tin capsule itself and/or solvents involved during the transfer of the analytes to the EA–IRMS system were tested to reveal the overall uncertainties associated with isotope analytical procedures. Our finding was that even a blank tin capsule (average weight = 65.80 mg) showed carbon contents of 61 ± 13 μgC with an amplitude of approximately 0.4 nA (Figure 2), while N content <0.1 nA was detected. Moreover, the tin capsule treated with 450 μL solvent (ACN, DCM and hexane) was used for transferring the target analyte from an extraction vial into a tin capsule. Subsequent drying for several hours in RT was consistent in providing the blank tin capsule with a peak height of approximately 0.4 nA (Figure 2). This indicates that N_2_-drying might effectively remove organic solvents before isotopic measurement. 

We assumed a very low carbon content in the blank tin capsules. To reveal the origin of unexpected carbon content, blank capsules were cleaned with DCM:MeOH (1:1, v:v) overnight, rinsed, and stored at 60 °C before use. Nonetheless, the cleaned blank tin showed a peak height of carbon <0.2 nA, which was much lower than the analytical blank (Figure 2). Overall, the unintended effect did not disappear completely during the isotope analysis. Similar to these results, previous studies showed that the interference effects of the blank tin capsules were consistent when using tins precleaned with an organic solvent mixture followed by combustion (400 °C, 5 h) [16,19]. Contamination may occur in the EA–IRMS system internally (e.g., from the gas line or autosampler) during isotope measurement. This indicates that the blank carbon content accounts for a non-negligible proportion of the analyte (Figure 2) and will not seriously contribute to isotope analysis. Thus, we suggested that blank correction should be considered when estimating the carbon contents of target compounds and δ^13^C calculation [18]. To minimize the unintended contamination related to tin capsule use, the washed tin was used for further analysis.

### 2.2. δ^13^C Variations in Standard Material: Before and after HPLC Purification

To validate the reliability of isotope information by HPLC/EA–IRMS, two aspects were considered in this study: (1) the relationship of amplitude (i.e., amount of analyte transferred to tin capsule) to the carbon isotope value and (2) the effects of HPLC performance on isotope variability. As shown in Figure 3, the δ^13^C of CHL before HPLC was −26.18 ± 0.09‰ (average ± SD, *n* = 27), which was approximately 0.85‰ different than that after the HPLC fraction of -27.03 ± 0.30‰ (*n* = 18). Slightly lower isotope values after HPLC were also observed for CYN. That is, the carbon isotope composition of CYN after the HPLC fraction was on average 0.44% lower (average ± SD: −26.19 ± 0.65‰, *n* = 21) than that before the HPLC fraction (average ± SD: −25.75 ± 0.08‰, *n* = 24). 

This isotopic difference might be related to the amplitude (or injection amount) of our target pesticides (Figure 3). The amplitude range of the CHL standard in our study was from 2.31 nA to 13.38 nA (the corresponding weights of standard material per tin capsule ranged from 0.01 mg to 0.13 mg), which was much broader than the amplitude from 1.52 nA to 2.48 nA (the estimated weight 0.03 mg to 0.08 mg) of CHL standard after the HPLC fraction. Thus, the amplitude of CHL from the HPLC fraction was close to the lowest level of analyte injection weight (mg) of powder CHL. The CYN standard results showed amplitudes per tin capsule from 4.32 nA to 14.15 nA (corresponding weight of the standard material from 0.01 mg to 0.15 mg), while the amplitudes were from 1.50 nA to 5.08 nA for CYN after the HPLC fraction (corresponding to their estimated weight from 0.03 mg to 0.15 mg per tin capsule). Moreover, SD was slightly different before and after the HPLC fraction (CHL: 0.09 vs. 0.30; CYN: 0.08 vs. 0.65). Higher precision in EA–IRMS relative to HPLC/EA–IRMS was also reported in other studies using other organic compounds, e.g., 0.08 vs. 0.16 in phenylalanine 11. Such isotope variability dependent on the sample amount (amplitude by IRMS) was widely reported in IRMS platforms. For instance, metabolite desphenylchloridazon formed by degradation of the herbicide chloridazon showed at least a 1‰ difference in δ^13^C values depending on the injection amount of C, while a larger standard deviation is reported with a smaller injection amount [13]. The overall result confirms that HPLC performance after solvent removal by N_2_-drying did not significantly change within a 1‰ difference for isotope measurements. To reduce the analytic errors from our EA–IRMS system, we also suggest a detection requirement of analyte per tin capsule (amplitude > 3 nA) in the HPLC/EA–IRMS method. Ultimately, securing the appropriate analyte amount in a tin capsule (approximately >0.03 mg) is important to produce reliable isotope compositions of specific compounds present in the environment by HPLC/EA–IRMS applications.

### 2.3. Application of HPLC/EA–IRMS and δ^13^C Determination of Target Compounds in a Soil-Crop System

The pre-treatment procedure in samples of interest for CYN and CHL detection traditionally is known to follow solid phase extraction (SPE) based on silica [24]. However, our pre-treatment procedure involved SPE extraction combined with HPLC separation (see Section 3). To evaluate the needs of the complex pre-treatment procedures, the analytic procedure was applied to a CYN-based product (11 mg/mL), a CHL-based product (9 mg/mL), and pesticide-free samples (topsoil and plant parts from a local market). Briefly, HPLC chromatograms showed that pesticides were present for CHL at 8 min and for CYN at 12 min under 45% ACN as a mobile phase, at least 2 min from the peaks of the pesticide-free sample matrix (interfering substances, present < 3.5 min) (Figure 4). This suggests that single SPE extraction (generally adopted for pesticide quantification) would experience unintended contamination from sample matrix effects when isotope values of pesticides are measured by an EA–IRMS system due to a lack of chromatographic separation ability. Consequently, SPE extraction after HPLC performance for collecting pesticide fractions would be essential to effectively exclude the interfering compound (‘matrix’ effect) and isolate residual pesticides, particularly from crop samples. 

The percentage of CHL in soil decreased to 50.75 ± 0.02% (amount estimated as 0.25 ± 0.03 mg/g) and that of CYN decreased to 80.13 ± 1.22% (estimated concentration 0.80 mg/g). In contrast to the concentration-based results (Figure 5), there was no significant change in δ^13^C in pesticides from soils, only a slight increase in δ^13^C (on average 1.7‰ for CHL from the soil) at 45 days relative to the initial time (−29.90 ± 0.02‰, Table 1). 

Moreover, CYN extracted from the soil at 0d was −29.41± 0.27‰, which changed slightly within <1.0‰ over 45 days (Table 1). Overall, the δ^13^C values in CHL and CYN extracted from pesticide-treated soil were not significantly changed at 45 d (on average δ^13^C < 1.7‰ with overlapping SD of analytical uncertainty). The absence of significant isotope changes might be less influenced by biodegradation, which is assumed to induce significant isotope changes in pesticides based on kinetic isotope fractionation (e.g., [7]). Like our results, other studies reported that pesticide δ^13^C compositions from sterilized soil (less dominated by microbial activity) showed a consistent pattern (on average δ^13^C < 1.0‰) for fenopropathrin, deltamethrin, α-cypermethrin [9], and lambda-cyhalothrin [8] under GC–IRMS. Even unsterilized soil (assuming active microbial biodegradation) showed a δ^13^C increase of approximately <2‰ within 40 days [8,9]. Such a negligible isotope change was also reported in a short-term lab-scale experiment (similar to 45 days in this study) in other pesticides such as butachlor, S-metachlor, and metalaxyl as analyzed by GC–IRMS [7]. Other studies report that δ^13^C did not change significantly for different pesticide amounts (2 mg/kg vs. 10 mg/kg) [9], planted or unplanted mesocosms, or soil types (forest soil vs. vineyard mesocosm) [7]. This indicates that degradation could trigger small, stable isotope changes, particularly for CHL and CYN in a soil environment. Other processes induce non-significant isotope fractionation such as sorption and leaching [6,8], which could be the major behavior of pesticides in the environment. The estimation of carbon isotope changes of CHL and CYN helped to broaden CSIA applications to monitor the short-term behavior of residual pesticides in environmental applications.

CHL and CYN residues from plants, transplanted to pesticide-treated soil for the experiment but previously grown in pesticide-free soil, were much less than those from soils (Figure 5). Indeed, only <2% of the initial pesticide treatment was detected in the plant grown in the CHL- and CYN-treated soils after 45 days, respectively. However, there were significant amounts (more than 55% of the initial treatment) of pesticides in top soils. These results suggest the pesticides might not persist well in crops. The residual CHL and CYN detected from plants suggest plant root uptake and transport of these pesticides, although their amounts in plants tended to decrease through the cultivation period. Similarly, the distribution of insecticides in plant parts decreased with an increased pesticide exposure period [1,25,26]. Although our study did not focus on the metabolites of CHL and CYN, there are reports that the amount of metabolites relative to the parental compound (CYN) increased [1,27]. The behavior may depend on plant tissue type, e.g., being highly abundant in leaves relative to fruits and flowers in tomato plants [1].

Compared to concentration changes, the overall CYN δ^13^C value throughout the sampling period from the leaf part (−28.70‰) was similar to that from the soil (−28.88‰) (Table 1). Additionally, overall CHL δ^13^C was 0.65‰ lighter in the plant than soil throughout the sampling period. In particular, the CHL δ^13^C difference in the plant and soil was on average from 1.7‰ to 2.9‰. The weak carbon isotope fractionation in our target substances might be related to the transformation process of an insecticide (parent compound) to its metabolites. For instance, CYN has a structure very close to its metabolite IN-J9Z38 formed by ring closure, which is frequently formed as a result of environmental degradation or plant metabolism [28]. Such transformation processes may not involve chemical bond breakage, leading to a weak carbon isotope fractionation in our target pesticides. The less variable isotopic patterns from CHL and CYN suggest that carbon isotope ratios can be used as fingerprints to distinguish contamination sources in chemical products, particularly in environmental samples such as soil and groundwater [12,29,30,31,32]. 

CSIA approach of pesticide CHL and CYN was first addressed in our small-scale indoor incubation experiments. As long as chemicals of interest are extracted from the soil or crop samples, the pesticide stable isotope approach helps distinguish the source of the pesticide(s) released from direct pesticides used and/or unintended effects of residual pesticides in the environment. This is due to a negligible isotopic change in the pesticides in the environment within a short timeframe. However, the procedure for transferring the analyte and removing any solvents in the tin capsule might result in analyte loss and decrease the amplitude detected in the EA–IRMS system compared to the actual powder weight. In fact, when the commonly used CSIA system is employed directly via on-column injection in GC–IRMS, the analyte amount is <1 microgram of glyphosate [31] or <135 nmol C of desphenylchloridazon [13], which is demanding significantly lower than that in our EA–IRMS approach. Thus, the sensitivity-improved EA–IRMS system, referred to as nano-EA–IRMS [16], and highly effective preparation steps for extracting a large sample amount may improve HPLC/EA–IRMS applications for a wide range of pesticides.

Although the detection limit for our EA–IRMS system was not good enough to analyze other elements (such as ^2^H,^15^N and ^37^Cl, [5]) in pesticides, the EA–IRMS-based method has potential advantages if the target compound is well isolated, purified, and collected beyond the detection limit. This is because the EA–IRMS platform is not affected by the molecular characteristics of organic compounds. In this regard, EA–IRMS is a reasonable tool to access chemical compounds with diverse polar/high molecular weight and their metabolites that are often more persistent and polar than their parental compounds in the environment [13]. Indeed, LC–IRMS can be used for a polar compound only but cannot be used for N isotope analysis, and GC–IRMS can be applied to midpolar or apolar compounds only but provides C and N in separated runs. Rather, EA–IRMS provides isotopic compositions of multi-elements (i.e., C and N) simultaneously in about 11 min. Additionally, EA–IRMS instrumentation is not expensive and is common in a stable isotope facility lab, and analytic services involving this technique are widely available. Multi-element CSIA leads to enhancing the discrimination power to verify sources of pesticide pollution in environments, rather than CSIA based on carbon only, as chloridazon standards are distinguished among suppliers [13]. Therefore, more analytic efforts should be involved in making more sensitive EA connected to IRMS (e.g., [19]) to produce dual isotopes more reliably in pesticides. 

In conclusion, our HPLC/EA–IRMS approach was applied to improve the CSIA application availability for the pesticides, CYN and CHL with high polarity and molecular weight (Figure 1). Moreover, the HPLC performance with the solid phase extraction procedure reduces effectively the interference effects from sample matrices such as soil and crop samples but is also capable of obtaining reliable isotope measurements in residual pesticides. Overall, negligible changes in the carbon isotope value in pesticides propose that the pesticide remaining in the soil is directly related to the pesticide product applied to agricultural environments. Therefore, CSIA might successfully uncover the primary source of pesticides when severe CHL and CYN contamination events occur in the field.

## 3. Materials and Methods

### 3.1. Chemicals and Reagents

The analytical standards cyantraniliprole (CYN) and chlorantraniliprole (CHL) (>98% purity) were purchased from FUJIFILM Wako Pure Chemical Corporation (Japan). Acetonitrile, ethyl acetate, dichloromethane, and hexane, all HPLC grade, were supplied by Merck (Darmstadt, Germany). Anhydrous sodium sulfate (reagent grade: >97%) and NaCl were purchased from Sigma-Aldrich (St. Louis, MO, USA) and Junsei (Tokyo, Japan). Deionized water was prepared using an Aquapuri 5 series system with a resistivity level 18.2 MΩ cm (Young In Chromass, Republic of Korea). Pesticide products of CHL (5%, Altacoa^®^, FarmHannong, Kyungju, Republic of Korea) and CYN (10.26%, Benevia^®^, FarmHannong, Kyungju, Republic of Korea) were purchased from a local agricultural market in Republic of Korea.

### 3.2. Validation of HPLC/EA–IRMS Isotopic Measurement

To investigate possible bias during HPLC purification and subsequent pre-concentrating processes, isotopic compositions of pesticide standards before and after HPLC chromatographic separation were compared. Before HPLC, CYN and CHL standards from 0.01 mg to 0.15 mg were prepared in tin capsules (pressed capsule, 10 by 10 mm, Elemental Microanalysis, UK). After HPLC, the CYN and CHL were respectively dissolved in 45% and 55% acetonitrile (ACN, HPLC grade, purity > 99.9%) in deionized water (approximately 2.1 mg/mL). Then, 50 μL of dissolved standard substance was injected into the HPLC. Chromatographic separation was carried out by Zorbax Eclips XDB C18 column (Agilent) at 30 °C. The mobile phases at a flow rate 1 mL/min were 45% ACN for CYN and 55% ACN for CHL in Table 2. CYN and CHL eluted around 8 min and 6 min, respectively, and were fraction-collected. The HPLC system Agilent 1260 Infinity II series (Agilent, Santa Clara, CA) consisted of a quaternary pump (G7111B; Agilent), a column temperature controller (G7116A; Agilent), an autosampler (G7129A; Agilent), autosampler thermostat (G1330B; Agilent), an online-photodiode-array detector (DAD; G7115A; Agilent), and a fraction collector (G1364F; Agilent). The fractions were pooled and prepared from 0.4 mL to 1.5 mL in duplicate to determine the detection limit of our HPLC/EA–IRMS method. 

Subsequently, the fractions were dried under nitrogen gas, re-dissolved in hexane and DCM, filtered using a glass fiber GF-5 filter (estimated to 0.03mg to 0.15mg) and transferred to tin capsules. To avoid effect of washed solvents, the tin capsules were dried until they reached a constant weight (approximately >5 h at room temperature), and they were then introduced into EA–IRMS for isotopic measurements. To reduce putative contamination effects from the tin capsule itself [16,19], tin capsules were soaked with DCM:MeOH (1:1, v:v) overnight, rinsed and kept at 60 °C before use. The EA–IRMS system is composed of EA (Elementar Vario Isotope Select, Elementar, UK) coupled with IRMS (Isoprime vision, Elementar, UK). The EA–IRMS diluter function was not used during isotope measurement. Carbon isotope values are reported in per mil (‰) using conventional delta notation relative to the international standards Vienna PeeDee Belemnite (V-PDB) and air, respectively:(1)$$\delta \;X\;\left(‰\right)=\left(\;\frac{R_{sample}}{R_{standard}}\;-\;1\;\right)\times 10^{3}\;$$
where *R* denotes the ^13^C/^12^C ratio for carbon. Blank corrections using an analytical blank were applied for every δ^13^C calculation. All reported isotope ratios are expressed as arithmetic means of replicate measurements with standard deviation in δ^13^C. Analytic precision was assessed as the SD of laboratory working standards, which were within 0.2 ‰ for δ^13^C analysis.

### 3.3. Pesticide Application and Plant Growth 

To illustrate the potential of the HPLC/EA–IRMS method in agricultural measurements, indoor container experiments were set up from mid-April 2021 to mid-June 2021 in the Isotope Ecology and Environmental Science Laboratory at Hanyang University. The containers were plastic and cuboid planters (52 cm × 14 cm × 15 cm) commonly used for home gardening. Potting soil mix was purchased from a local market (Ildungsangto^®^, Tosung, Republic of Korea) and was composed of 65–70% cocopeat, 10–15% zeolite, 5–10% perlite, and 1–4% biochar. The pH was 7.0, and the relative humidity was approximately 45%. Each container was filled with 4 kg (oven dry weight 2.3 kg) of the potting soil and reached 10 cm in depth. In total, 10 containers were prepared and allocated to either CYN or CHL treatment. Each pesticide product was mixed with 4L of water in a bucket and poured into the base of the container, allowing the container soil to absorb it. Final concentrations of CYN and CHL in the spiked soil container (pesticide mg/ soil kg) were 250mg/kg and 500mg/kg, respectively. This high concentration is not a recommended guideline amount for general pesticide users in Republic of Korea but was used for obtaining crop as well as soil samples with large amount of pesticide residue to reveal the unintended pesticide contamination sources in soil as well as crops. 

Two days after pesticide treatment, fast-growing leafy vegetable lettuces (approximately one month old after seedling) that were purchased from a local farmer’s market were transplanted into pesticide-treated containers. We assumed the pesticide was distributed evenly through soil in the container. Six plants were planted in each container, and 30 total plants were raised for each pesticide. The cuboid planters were incubated for two months in front of a sunroom-type window, and 4L of water every week were supplied to a base of the container. Then, the soil and crop samples (leaf parts from one plant) were collected at 10, 20, 30 and 45 days. Next, 150 g (wet weight) of soil was sampled with plastic spoons, and harvested leaves were rinsed with tap water to remove surface dust, chopped into small pieces, and then placed into polyethylene bags. Samples were immediately frozen and stored in a −20 °C freezer until analysis. 

For two months, the average temperature was 19 °C with a maximum of 26 °C and a minimum of 15 °C. Additional light was not used in this study to maintain natural day and night conditions. Further, other external factors (e.g., rainfall) were not considered as the experiment was conducted under indoor conditions.

### 3.4. Sample Preparation: Isolation and Purification of Residual Pesticides In Situ Samples

The extraction method was based on previous protocols [23,24] with a slight modification of solvent volume. The overall workflow to extract residual pesticides from soil and crops is summarized in Figure 6. Samples were homogenized and extracted with 25 mL of ACN in a falcon tube and were centrifuged at 2500 rpm for 5 min. The supernatant was filtered through a syringe membrane filter (0.2 μm, PTEE-H). This ACN extraction procedure was conducted twice. The supernatant was transferred to a glass tube and concentrated to <1 mL in a turbo nitrogen evaporator with a water bath temperature of 40 °C.

Further, the liquid–liquid partitioning process was different for CYN and CHL. For CYN, 20 mL distilled water, 5 mL saturated NaCl solution, and 20 mL hexane were added to the concentrated supernatants in the glass tube, and the glass tube was vigorously shaken. After complete layer separation, the lower layer was passed through 5 g anhydrous sodium sulfate. Then, the extract was collected and concentrated to dryness using a turbo nitrogen evaporator. The residue was dissolved in 3 mL DCM. The concentrated extract was loaded into a silica cartridge (6 cc Bond Elut, Agilent Technologies), which was pre-activated with 10 mL DCM. Then, the solution was washed with 3 mL 10% ethyl acetate in DCM. CYN was eluted with 3 mL 40% ethyl acetate in DCM. The eluted fraction was concentrated using a nitrogen evaporator, and the residue was dissolved in 2 mL 45% ACN in DW. 

For extracting CHL, 10 mL distilled water, 10 mL saturated NaCl solution, and 20 mL DCM were added to the concentrated supernatant in a glass tube. After waiting until the layers were completely separated, the lower part was drained into 1.5 g anhydrous sodium sulfate. This partitioning was repeated with 10 mL DCM. Then, extracts were collected, pooled, and concentrated to dryness using a turbo nitrogen evaporator. The residue part was re-dissolved to 5 mL 20% ethyl acetate in hexane. The re-dissolved residue was loaded into a silica cartridge (1 g, 6 cc Bond Elut, Agilent Technologies), which was pre-washed using 10 mL hexane. Then, the cartridge was eluted with 6 mL 30% ethyl acetate in hexane and 6 mL 50% ethyl acetate in hexane. The eluted fractions were pooled and dried using a nitrogen evaporator, and then the residue was dissolved in 2 mL 55% ACN in DW and was stored at −20 °C prior to the HPLC procedure. 

Then, pesticide extracts (<100 μL from soil and <900 μL from lettuce samples) were injected for HPLC. HPLC purification procedures were conducted with 10–12 repetitions to obtain pesticide fractions of CYN and CHL. The collected fractions were pooled and prepared in triplicate as described previously.

### 3.5. Standard Calibration Curve and Estimating Pesticide Amount 

Standards of CHL and CYN were prepared by dissolving 20 mg of the compound in 20 mL solvent to obtain a 100 mg/L stock solution, respectively. From this stock solution, a working standard solution (20 mg/L) was prepared by dilution in solvent. This was then serially diluted to obtain standard solutions of 0.01, 0.05, 0.10, 0.20, 0.50, 1.00, 2.00, 3.00, 5.00, and 10.00 mg/L. An aliquot of 2.0 μL was injected into the HPLC, and a standard calibration curve was prepared based on the peak area. Limit of quantification (LOQ) was 0.082 mg/L for CHL and 0.0109 mg/L for CYN in our HPLC systems. LOQ was calculated as: LOQ (mg/ kg) = [minimum detectable amount (ng)/injection volume (uL)] × [final sample volume (mL)/sample amount (g)]. The instrumental conditions are shown in Table 1.

## Figures and Tables

**Figure 1 molecules-27-08587-f001:**
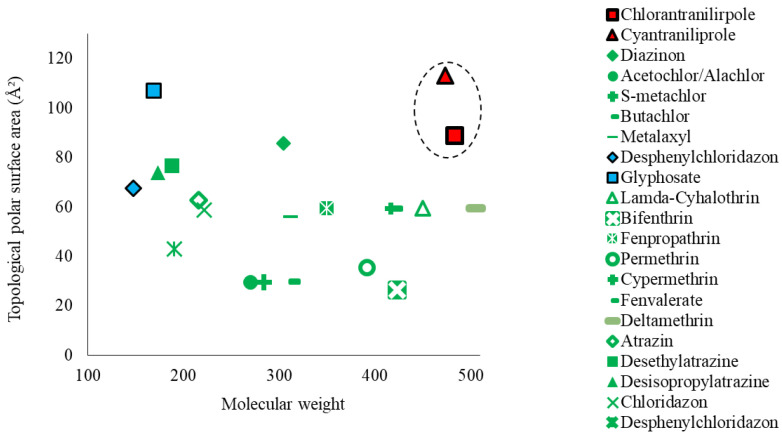
Distribution of diverse pesticides chemicals based on molecular characteristics and their analytic equipment, such as LC (or GC)–IRMS with blue symbols, GC–IRMS with green symbols [7,8,9,10,11,12,13], and EA–IRMS with red symbols (in this study) for stable isotope analysis.

**Figure 2 molecules-27-08587-f002:**
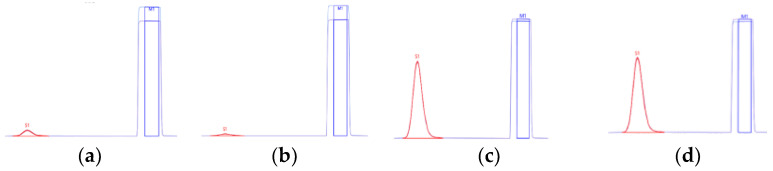
Chromatograms of blank (empty) tin capsule (**a**), MeOH/DCM prewashed tin capsule (**b**), and standard chlorantraniliprole (**c**) and standard cyantraniliprole (**d**) in EA–IRMS. Square-shaped peaks indicate reference CO_2_.

**Figure 3 molecules-27-08587-f003:**
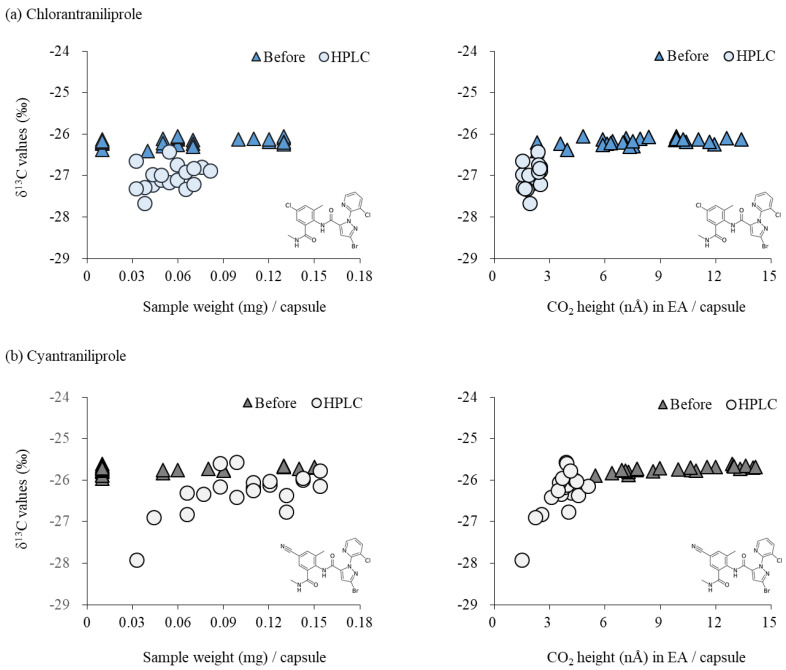
Effects of HPLC analytic procedure on determining isotope measurement in CHL standard (**a**) and CYN standard (**b**).

**Figure 4 molecules-27-08587-f004:**
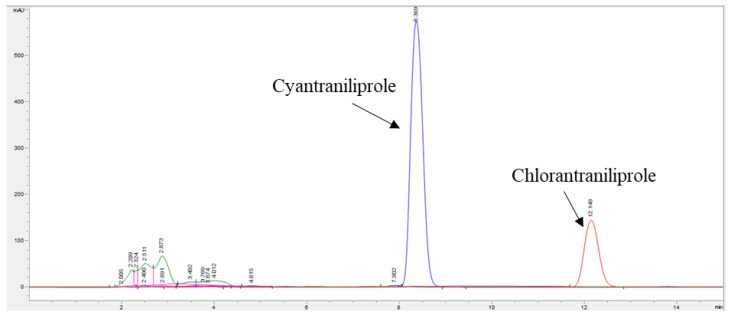
Chromatograms overlay in HPLC of two pesticides standards, cyantraniliprole (at 8.4 min) and chlorantraniliprole (12.3 min), and other pesticide-free samples of soil, leafy parts of lettuce, and root parts metrics (before 6 min) with constant flow (1mL/min) of 45% Acetonitrile in DW as mobile phase.

**Figure 5 molecules-27-08587-f005:**
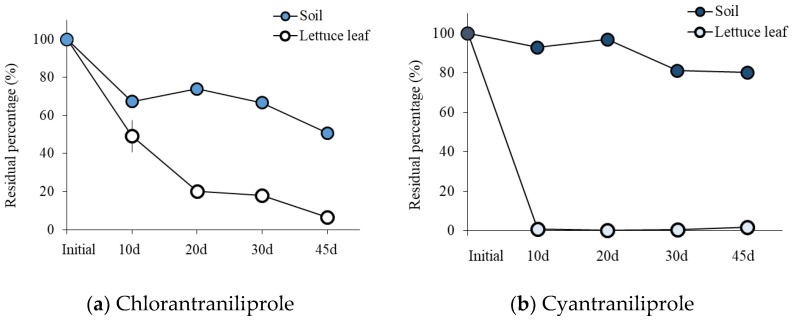
Estimate of amounts of residual pesticides in soil and lettuce leaves from indoor crops over a 45-day culture experiment (N = 3).

**Figure 6 molecules-27-08587-f006:**
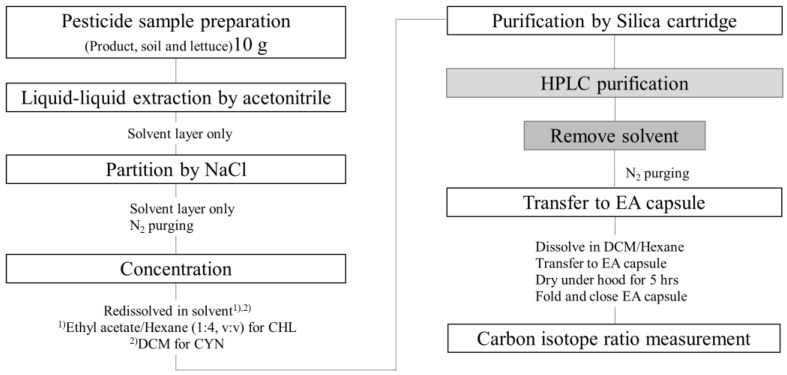
Schematic workflow to determine residual pesticides in samples for environmental and agricultural applications.

**Table 1 molecules-27-08587-t001:** Carbon isotope variability of chlorantraniliprole and cyantraniliprole extracted from samples.

	Soil				Crop		
	δ^13^C	Estimated C (mg/capsule)	Amount (mg/g)	N	δ^13^C	Estimated C (mg/capsule)	N
(a) *Chlorantraniliprole*							
Initial	−29.90 ± 0.02	0.176 ± 0.010	0.5 *	2	Not applied		
10d	−29.24 ± 0.17	0.171 ± 0.004	0.337	3	−30.35 ± 0.06	0.268 ± 0.032	3
20d	−28.37 ± 0.13	0.222 ± 0.031	0.370	3	−30.27 ± 0.50	0.206 ± 0.017	3
30d	−28.85 ± 0.10	0.093 ± 0.012	0.334	3	−30.56 ± 0.40	0.085 ± 0.010	5
45d	−28.24 ± 0.28	0.110 ± 0.018	0.254	3	−31.11 ± 0.10	0.130 ± 0.037	3
Overall	−28.92	0.154			−30.57		
(b) *Cyantraniliprole*							
Initial	−29.42 ± 0.27	0.159 ± 0.023	1 *	3	Not applied		1
10d	−28.74 ± 0.17	0.212 ± 0.019	0.93	3	−29.03	0.043	1
20d	−29.15 ± 0.05	0.173 ± 0.051	0.97	3	−28.79	0.042	1
30d	−28.28 ± 0.58	0.142 ± 0.013	0.81	3	−28.54	0.070	1
45d	−28.80 ± 0.56	0.143 ± 0.026	0.80	3	−28.45	0.047	1
Overall	−28.88	0.166			−28.70		

* Amount of pesticides exposed to soil at initial time.

**Table 2 molecules-27-08587-t002:** Carbon isotope variability of chlorantraniliprole and cyantraniliprole extracted from samples.

HPLC System	Cyantraniliprole	Chlorantraniliprole
Column	Zorbax Eclips XDB C18 column (4.6 × 250 mm, 5 µm)	Zorbax Eclips XDB C18 column (4.6 × 250 mm, 5 µm)
Mobile phase (duration time)	45% Acetonitrile in water (15 min)	55% Acetonitrile in water (15 min)
Flow rate	1 mL/min	1 mL/min
Column temperature	30 °C	30 °C
Detection	264 nm	254 nm
Injected volume	0.05mL to 0.50 mL	0.05mL to 0.50 mL

## Data Availability

Not applicable.

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
