# Peer review of "Stable Isotope Analysis of Residual Pesticides via High Performance Liquid Chromatography and Elemental Analyzer–Isotope Ratio Mass Spectrometry"

_molecules, 2022, doi:10.3390/molecules27238587_

Round 1

Reviewer 1 Report

Well written paper, and i have no major comments.

two minor comments:

1. p3

Moreover, traditional EA-IRMS systems require much higher amounts of analytes that produce reliable isotope variables compared to a GC-IRMS system in which analyte mass for EA-IRMS is approximately 20 μ gC (or 50 μ gN) [14, 16, 17], which is 1000 times larger than those for
GC-IRMS (e.g., around nanograms on column)

This sentence need to be clarified, and should be simplified right now its difficult to understand what you mean and slightly confusing for the reader.

2.

i dont see that you discuss this paper (Höhener, P., Guers, D., Malleret, L. et al. Multi-elemental compound-specific isotope analysis of pesticides for source identification and monitoring of degradation in soil: a review. Environ Chem Lett (2022). https://doi.org/10.1007/s10311-022-01489-8), please set your results in realtion to this,

Author Response

Point 1: “Moreover, traditional EA-IRMS systems require much higher amounts of analytes that produce reliable isotope variables compared to a GC-IRMS system in which analyte mass for EA-IRMS is approximately 20 µgC (or 50 µgN) [14, 16, 17], which is 1000 times larger than those for GC-IRMS (e.g., around nanograms on column).” This sentence need to be clarified, and should be simplified.. right now it is difficult to understand what you mean and slightly confusing for the reader.

RESPONSE: The sentence is re-written as “Moreover, traditional EA-IRMS systems require much higher amounts of analytes that produce reliable isotope variables compared to a GC-IRMS system. That is, analyte mass required for EA-IRMS is approximately 20 µgC (or 50 µgN) [14, 16, 17], while analyte mass for GC-IRMS via directly injection on column is around 0.2 to 0.02 µgC (or 0.5 to 0.05 µgN).”

Point 2: I don’t see that you discuss this paper - Höhener, P.; Guers, D.; Malleret, L.; Boukaroum, O.; Martin-Laurent, F.; Masbou, J.; Payraudeau, S.; Imfeld, G., Multi-elemental compound-specific isotope analysis of pesticides for source identification and monitoring of degradation in soil: a review. Environmental Chemistry Letters (2022). https://doi.org/10.1007/s10311-022-01489-8., please set your results in relation to this.

RESPONSE: We updated our reference list with the paper that review said, and cited throughout our manuscript. Please find our revision in the reference as ref. 5.

Reviewer 2 Report

A work entitled "Stable isotope analysis of residual pesticides via high performance liquid chromatography and elemental analyzer-isotope ratio mass spectrometry" by Hee Young Yun et Co. is interesting work, based on determination pesticide residues in samples. Pesticides are used to prevent crop damage by pest insects and pathogens and to prolong storage lives of agricultural products. But due to their chemical and toxicological profile could have negative impact human health and environment. 

There is a lot of techniques for determination of pesticide residues and risk assessment. Authors are interested on compound-specific stable isotope analysis - there is selectively accessible for several pesticides and pesticide metabolites. 

In my opinion The Authors should describe details about analytical technique - chromatographic conditions are described, but there is a lack of injected volume and/or sample preparation techniques.

In my opinion there is a lack of conclusion - what conclusions the authors have from their work and research?

Author Response

Point1: In my opinion, the authors should describe details about analytical technique-chromatographic conditions are described, but there is a lack of injected volume and/or sample preparation techniques.

RESPONSE: We added the information showing injection volume during HPLC performance in Table 2 and we revise the solvent types in supplementary info, which is Figure 6 in the revised manuscript.

Point2: There is lack of conclusion- what conclusions the authors have from their work and research?

RESPONSE: Conclusion part is added as “In conclusion, our HPLC/EA-IRMS approach was applied to improve the CSIA application availability for pesticides with a high polarity and molecular weight, CYN and CHL (Fig. 1). Moreover, the HPLC performance with the solid phase extraction procedure reduces effectively the interference effects from sample matrices such as soil and crop samples, but also is capable of obtaining the reliable isotope measurement in residual pesticides. Overall, negligible changes of carbon isotope value in pesticides propose that the pesticide remained in soil directly is related to the pesticide product applied to agricultural environments. Therefore, CSIA might successfully uncover the primary source of pesticides, when severe CHL and CYN contamination events occur in field.”

Round 2

Reviewer 2 Report

Once the manuscript has been revised by the authors - I have no comments and am of the opinion that it can be published in a journal